# Microstructure and Mechanical Properties of Nanocrystalline Al-Zn-Mg-Cu Alloy Prepared by Mechanical Alloying and Spark Plasma Sintering

**DOI:** 10.3390/ma12081255

**Published:** 2019-04-16

**Authors:** Jingfan Cheng, Qizhou Cai, Bingyi Zhao, Songfeng Yang, Fei Chen, Bing Li

**Affiliations:** 1State Key Laboratory of Materials Processing and Die & Mould Technology, Huazhong University of Science and Technology, Wuhan 430074, China; D201377244@hust.edu.cn (J.C.); D201577254@hust.edu.cn (B.Z.); M201770935@hust.edu.cn (S.Y.); M201770936@hust.edu.cn (F.C.); D201477244@hust.edu.cn (B.L.); 2School of Mechanical Engineering, Wuhan Vocational College of Software and Engineering, Wuhan 430205, China

**Keywords:** mechanical alloying and spark plasma sintering (MA-SPS), nanocrystalline Al-Zn-Mg-Cu bulk alloy, microstructure, mechanical properties

## Abstract

In this study, Al, Zn, Mg and Cu elemental metal powders were chosen as the raw powders. The nanocrystalline Al-7Zn-2.5Mg-2.5Cu bulk alloy was prepared by mechanical alloying and spark plasma sintering. The effect of milling time on the morphology and crystal structure was investigated, as well as the microstructure and mechanical properties of the sintered samples. The results show that Zn, Mg and Cu alloy elements gradually dissolved in α-Al with the extension of ball milling time. The morphology of the ball-milled Al powder exhibited flaking, crushing and welding. When the ball milling time was 30 h, the powder particle size was 2–5 μm. The α-Al grain size was 23.2 nm. The lattice distortion was 0.156% causing by the solid solution of the metal atoms. The grain size of ball-milled powder grew during the spark plasma sintering process. The grain size of α-Al increased from 23.2 nm in the powder to 53.5 nm in the sintered sample during the sintering process after 30 h of ball milling. At the same time, the bulk alloy precipitated micron-sized Al_2_Cu and nano-sized MgZn_2_ in the α-Al crystal. With the extension of ball milling time, the compression strength, yield strength and Vickers hardness of spark plasma sintering (SPS) samples increased, while the engineering strain decreased. The compression strength, engineering strain and Vickers hardness of sintered samples prepared by 30 h milled powder were ~908 MPa, ~8.1% and ~235 HV, respectively. The high strength of the nanocrystalline Al-7Zn-2.5Mg-2.5Cu bulk alloy was attributed to fine-grained strengthening, dislocation strengthening and Orowan strengthening due to the precipitated second phase particles.

## 1. Introduction

Al-Zn-Mg-Cu alloy is widely used in aerospace and transportation due to the advantages of high specific strength and superior forming properties [1,2]. The plates and bars of Al-Zn-Mg-Cu alloy prepared by traditional processes such as casting and plastic deformation have inferior mechanical properties because of coarse microstructure and uneven distribution of the second phase [3,4]. Recently, it is reported that injection molding [5] and powder metallurgy processes [6,7] can significantly improve the mechanical properties of the alloys. However, the grain sizes of the alloys obtained from these methods are all on the scale of micrometers. The latest studies [8,9,10] indicate that when the grain size of aluminum alloy is reduced to nano-size, the mechanical properties are significantly improved. Thus, nanocrystallization of aluminum alloys has attracted more and more research interests in recent years.

Mechanical alloying (MA) is a novel technology to prepare nanocrystalline alloy powders [11]. The immiscible components at room temperature can form supersaturated solid solution by MA. The formation of supersaturated solid solution can be attributed to the cold welding and fracturing of the powder particle during the long-term MA process [12,13]. In the subsequent step, densification methods such as hot pressing (HP), hot isostatic pressing (HIP) and spark plasma sintering (SPS) can be used to prepare bulk alloy from powders [14]. HP and HIP have limited ability to keep a fine microstructure because of the struggle between grain growth and densification kinetics due to long sintering cycles [15]. Dense bulk alloy can be obtained effectively by SPS process at lower temperatures and shorter sintering time. By using this process, the growth of powder grains is restrained, and the mechanical properties of alloy are significantly improved [16,17,18,19,20].

Some researches indicated that nanocrystalline bulk aluminum alloy prepared by the MA-SPS method have excellent mechanical properties. Gu et al. [21] used the MA-SPS method to prepare an Al-5wt.%Fe bulk alloy with a grain size of 32 nm. The yield strength, hardness and engineering strain were 1018 MPa, 1.61 GPa and 14.5%, respectively. Kong et al. [22] prepared Al-40wt.% Cu block alloy with a grain size of 200 nm by MA-SPS. The Vickers hardness of the alloy achieved 210 HV. Park et al. [23] prepared Al-47wt.%Mg block alloy by MA-SPS. The grain size of Al achieved 2.58 nm. The Vickers hardness of the alloy was 189 HV, which was five times that of solid Mg and seven times that of solid Al. However, the above studies were all limited to the binary Al alloys. There multi-component nanocrystalline bulk aluminum alloys prepared by MA-SPS were rarely reported.

In this study, nanocrystalline bulk Al-Zn-Mg-Cu alloy was prepared by the MA-SPS method using elemental metallic powders. The solid solution process of alloying elements and the change of grain size in Al powder in the process of ball milling were investigated. The formation of Al matrix structure and the second phase precipitation process in the sintering process were studied. The effect of milling time on the properties of the SPS samples was investigated. The strengthening mechanisms of nanocrystalline bulk Al-Zn-Mg-Cu alloy were also discussed.

## 2. Materials and Methods

### 2.1. Materials Preparation

Figure 1 shows the microstructure of the Al, Mg, Zn and Cu raw powders prepared by nitrogen gas atomization. The Al powder is nearly spherical. The Mg powder possesses a sheet-like, irregular geometry. The Zn and Cu powders are both spherical. The purity and particle sizes are listed in Table 1.

The Al, Zn, Mg and Cu powders were firstly dried in a vacuum drying oven. Then, the powders were placed in a stainless-steel vacuum tank with portions of 88% Al, 7% Zn, 2.5% Cu, 2.5% Mg (wt.%) and 5 wt.% of ethanol was added as a process control agent (PCA). Mechanical milling of mixed powder was performed in a planetary ball mill QM-3SP4 (Nanjing University Instrument Factory, Nanjing, China) under Ar atmosphere at room temperature. The parameters of the planetary ball milling were set as follows. The ball-to-powder mass ratio was 20:1, the mass ratio of the diameter of the stainless-steel ball was Ø 10 mm:5 mm = 1:3, the rotation speed was 400 rpm, and it stopped for 10 min after 60 min of operation. The ball milling time was 5, 10, 15, 20, 25, 30 and 40 h. After milling, the ball-milled powder was dried in a vacuum oven at 70 °C for 1 h to evaporate the residual ethanol in the powder.

The milled powder after drying was put in a graphite mold with an inner diameter of 30 mm, and then was sintered in a LABOX-1575 SPS apparatus (SINTER LAND, Nagaoka, Japan). The sintering process was shown in Figure 2. The initial pressure was 10 MPa and the heating rate was 50 °C/min. When the temperature was raised to 200 °C, the pressure was increased to 50 MPa. When the sintering temperature reached 500 °C, the holding time was 3 min. As soon as the holding procedure terminated, the temperature was lowered at 50 °C/min. When the mold temperature dropped to 200 °C, the pressure was reduced from 50 MPa to 10 MPa. After sintering, the sample was unloaded and cooled down in the vacuum chamber. The cylindrical samples with a diameter of 30 mm and height of 10 mm were obtained after SPS processing. During the sintering process, the vacuum in the chamber was maintained at 5–20 Pa to avoid oxidation.

The different milling time powder samples were denoted by P5, P10, P15, P20, P25, P30 and P40, and the corresponding sintered samples are denoted by S5, S10, S15, S20, S25, S30 and S40, respectively.

### 2.2. Material Characterization and Performance Testing

The phase composition of the powders and sintered samples was analyzed using the X-ray diffraction (XRD) measurements (Cu K_α_, λ = 0.15406 nm, Shimadzu XRD-7000S, Kyoto, Japan) with a scanning speed of 10°/min, a scanning range of 10°–90° operating at a tube voltage of 40 kV and tube current of 40 mA. Based on the crystal plane angle and the half width of Al, the grain size, the full width at half maximum (FWHM) and the microscopic strain were calculated by the Williamson–Hall method [24].

Milled powders were hot mounted (180 °C, 2.5 min) in conductive resin with carbon filler. The compact samples of 5 × 5 × 5 mm^3^ in size were cut out of the SPS samples by the electro-discharge machining. All samples were metallographically ground and polished. After grinding and polishing, they were etched with Keller’s reagent (95% H_2_O, 2.5% HNO_3_, 1% HF, 1.5% HCl, vol.%) for 15 s. The microstructures of the powders and sintered samples were observed using a field-emission scanning electron microscope (FESEM, Nova Nano SEM 450, FEI, Eindhoven, Netherlands), and an EDS analysis of the phases was performed using energy-dispersive spectroscopy (EDS, Oxford X-Max 50, Eindhoven, Netherlands).

The SPS samples were observed at high magnification using a transmission scanning electron microscope (FETEM, Tecnai G2 F30, FEI). The preparation process of the transmission sample was as follows. First the sample was mechanically thinned to 50 μm and punched into a Ø 3 mm wafer. An MTP-1A magnetic drive double-spray electrolysis was thinned to the microporous level, where double-spray voltage was 18–20 V. The electrolyte was 25% HNO_3_ + 75% CH_3_OH (vol.%). The electrolyte was controlled to a temperature below −25 °C by liquid nitrogen during the thinning process.

A cylinder sample of Ø 6 mm × 10 mm was cut from the sintered samples by slow wire cutting. The direction of compressive load was parallel to that of the SPS load. The compression test should refer Metallic materials—Compression test method at room temperature (GB/T 7314-2005) [25] using a Shimadzu AG-100 KN machine at a speed of 0.2 mm/min. The Vickers microhardness test should refer to the Metallic materials—Vickers hardness test (GB/T 4340-2009) [26]. The Vickers microhardness of the SPS samples was measured with a load of 4.9035 N for a dwelling time of 15 s using 430 SVD Vickers microhardness machine and an average of ten indentations was reported.

## 3. Results and Discussion

### 3.1. Morphology and Grain Size of Milled Powders

Figure 3 shows the morphologies of the Al-7Zn-2.5Cu-2.5Mg milled powders for different milling times. The particle size first increases and then decreases with the milling time. There are five stages during the process of ball milling, such as flattening stage, welding stage, fracture stage, formation stage of equiaxed particle and dynamic equilibrium stage [27,28]. The P5 powder changed from the original spheres (Figure 1) to flakes (Figure 3a). The illustration of Figure 3a shows a noticeable crack in the flaky powder. After 10 h of ball milling, the flat powder was welded and formed a lamellar shape as indicated by the yellow arrows in Figure 3b. The P15 particles were clearly broken, and particles were formed irregular shapes with a wide size distribution, as shown in Figure 3c. The P20 particles were further broken into fine particles, and approached an equiaxed shape, as indicated by the red arrows in Figure 3d. The P25 particles were equiaxed and the size decreased further, as shown in Figure 3e. The P30 particle size distribution was consistent and concentrated around 2–5 μm, as shown in the histogram of Figure 3h. The particles size was smallest during the milling process, as shown in Figure 3f. When the ball milling time was extended to 40 h, the particle size increased slightly while the morphology remained equiaxed, as shown in Figure 3g. This phenomenon indicated that particles had been welded slightly. After 30 h of ball milling, the powder was in dynamic equilibrium.

Related studies [29,30] demonstrate that the powders are plastically deformed under the combined action of extrusion impact, and shear, caused by the continuous impact and grinding of the balls. As the milling time increased, the degree of the plastic deformation of the powders gradually increased, resulting in strain in the powders, which causes work hardening of the powders. The hardened powders were broken gradually during the ball milling. The reason for the decrease of particles size was that the crushing probability of powders became greater than the welding probability of the particles. By further extending the ball milling time, the broken particles welded again. It is evident that the mechanical alloying process was a repeated cycle of welding and fracture. After 30 h of ball milling, the powders were equiaxed and its morphology did not change significantly further. The cold welding and crushing of the powders achieved a dynamic equilibrium. 

Figure 4 shows the cross-section morphologies of the powder particles at different ball milling time. Figure 4a shows that the internal morphology of the particle was lamellar after 10 h of ball milling. The shape of P20 particles became equiaxed and the pores appeared inside the particles, as shown in Figure 4b. When the ball milling was continued to 30 h, the particles were equiaxed, and the internal pores were furtherly reduced, as shown in Figure 4c. This phenomenon further indicates that the pressing force of grinding balls increased the internal densification of the particles during the mechanical alloying process [31].

Figure 5 shows XRD patterns of the Al-7Zn-2.5Cu-2.5Mg powder at different ball milling times. Figure 5a shows that there is no new phase formed in milled powders. The intensity of the diffraction peaks of Al, Zn, Mg and Cu are gradually decreased and broadened as the ball milling time increases. After 30 h of ball milling, the diffraction peaks of Zn, Mg and Cu disappeared completely.

The room temperature solubility of Zn, Mg and Cu atoms in α-Al is very small [32]. At room temperature, the solubility of Cu in Al is less than 0.01 wt.% [33], the solubility of Zn in Al is less than 0.1 wt.% [34], and that of Mg in Al is less than 1.8 wt.% [35]. The XRD diffraction peaks of Figure 5a indicate that the mixed powder passed through a strong collision of the grinding balls with increasing milling time, and high-density dislocations were generated inside [36]. Numerous crystal defects provided a rapid channel for the inter-diffusion of Al, Zn, Mg and Cu atoms. The Zn, Mg and Cu atoms were solid-dissolved into the Al matrix to form a single alloy phase. This was a supersaturated solid solution α-Al phase.

The main peak of α-Al (111) crystal plane was selected to analyze the influence of the ball milling time on lattice parameter and lattice strain of α-Al crystal. Figure 5b shows the localized patterns from 37.6° to 39.6°. It indicates that Al peak broadened with the extension of ball milling time. The Al (111) peak shifts from 38.329° to 38.472° as the milling time increased from 5 h to 40 h. That is because the atomic radius of Zn (0.134 nm) and Cu (0.128 nm) are smaller than that of Al (0.143 nm). Although the atomic radius of Mg (0.160 nm) is larger than that of Al, the sum of the atomic fractions of Zn and Cu is larger than that of Mg. Therefore, the solid solution of Zn, Cu and Mg into α-Al causes the reduction of α-Al lattice constant, and the diffraction peaks moved toward a large angle.

Using the Williamson–Hall method (Equation (1)) [24,37], the Al diffraction peaks in Figure 5a are fitted to obtain the average grain size and microscopic strain of α-Al in the composite powder at different ball milling times:(1)βcosθ=0.9λ/d+4εsinθ
where *β* is the FWHM, *λ* = 0.154056 nm, *θ* is the Bragg diffraction angle, *d* is the average grain size, and *ε* is microscopic strain.

According to the standard Si sample, the diffraction pattern was corrected by Jade 6.0 software. The half-height width of the diffraction peak with the corresponding diffraction surface was obtained. The values of *θ* and *β* at the five diffractive faces (111), (200), (220), (311) and (222) of Al were selected, and a straight line for *β*cos *θ* and sin *θ* was obtained. The average grain size was calculated according to the intercept *c* of the straight line, *d* = *Kλ*/*c*. The average microscopic strain *ε* = *m*/4 was calculated according to the slope, *m* of the straight line, and the results were shown in Table 2. The average grain size decreased gradually as the milling time increased from 0 to 40 h. The minimum average grain size was 22.3 nm for P30, and the microscopic strain was 0.156%. The average grain size for the extended time of ball milling slightly increased to 23.0 nm at 40 h, while the microscopic strain was 0.138%. This indicated that the grain size and microscopic strain of the alloy powders tended to stabilization after ball milling for 30 h.

### 3.2. Microstructure of SPS Samples

#### 3.2.1. Microstructure of SPS Samples

Figure 6 shows the XRD patterns of SPS samples using different milling time powders. Comparing Figure 6 with Figure 5a, the intermetallic compound Al_2_Cu phase formed in the sintered sample. This is because the supersaturated solid solution of Cu in α-Al during mechanical alloying. During the SPS process, the thermal effect of the sintering caused the desolventizing of Cu and reacting with the Al matrix to form a second-phase Al_2_Cu.

The calculated average grain sizes using the Williamson–Hall method for SPS samples were shown in Table 3. Corresponding to the grain size of the powders in Table 2, the grains of the SPS samples grew significantly during sintering. However, the grain sizes of the SPS samples were maintained at the nanoscale. The S30 had a minimum grain size of 53.5 nm. The grain sizes of S5, S10 and S15 were greater than 100 nm, and the Williamson–Hall method was thus not applicable to them. Thus, they are not listed in Table 3.

Figure 7 shows the microstructure of the sintered samples for different ball milling times. Figure 7a shows distinct lamellar layers in the S10 sample. The morphology is consistent with that of the cross-sectional morphology of 10 h milled powders. S20 sintered sample (Figure 7b) was compact and had no powders structure characteristics, but some small holes appeared in the sample. This is because the 20 h milled powders appeared as a near sphere and the sintering was more adequate. As shown in Figure 7c, S30 sample had more compact structure than that of S20 sample, and the pores became smaller and dispersed due to the finer milled powders. This is beneficial for the performance of the sintered sample.

Based on the above test results, it can be known that the shape of the ball-milled powders influences the microstructure of sintered samples. When the shape of milled powders changes from flake to fine near sphere, the internal pores become smaller, which is more favorable for sintering.

Microstructure of S30 sample and energy-dispersive spectroscopy (EDS) analysis of secondary phase were shown in Figure 8. It can be seen from high magnification microstructure in Figure 8a, densified microstructure could be obtained by SPS using 30 h milled powders. The micron-sized second phase particles are distributed in the matrix and it can be identified as Al_2_Cu from EDS in Figure 8b. The formation of Al_2_Cu is due to reaction of supersaturation Cu atoms and its surrounding Al atoms during the SPS process.

Figure 9 shows a TEM micrograph of the S30 sample and a histogram of the size distribution of α-Al grains. As shown in Figure 9a, all grains distributed in the TEM image of S30 are nano-sized grains. To calculate their sizes, the morphology of the dispersed α-Al grains is marked with yellow dashed lines shown in Figure 9a. The grain size distribution histogram was obtained by Image Plus 6.0 software counting the crystal grains in more than twenty TEM micrographs, as shown in Figure 9b, each TEM micrograph marked more than one hundred grains. The distribution of the α-Al grain-size histogram in Figure 9b shows that the maximum grain size is 88.4 nm, and 91% of the grain sizes are 44 nm to 78 nm. Comparing with the grain size of 30 h ball milling powder in Table 1, the grain size of α-Al grew significantly during the SPS process. However, owing to the low sintering temperature and short time of the SPS, grain growth is suppressed, and the grain size of the sintered sample remained on the nanometer scale. This result is consistent with the results calculated in Table 3 using the Williamson–Hall method.

Figure 10 shows TEM bright-field micrograph, HRTEM micrograph and FFT image of MgZn_2_ nanoparticle. As can be seen from the Figure 10a, the precipitated phases with size of 30 nm to 50 nm were dispersed in α-Al matrix and the morphology of precipitated grain are polygonal (Figure 10b). 

From the inter-plane distance measurement (Figure 10c) and the Fast Fourier Transform (FFT) image (Figure 10d), the precipitated phase was identified as MgZn_2_ (*η*). In this study, owing to the rapid sintering technique of SPS, the particle size of the second reinforcing-phase MgZn_2_ was reduced to less than 50 nm. 

#### 3.2.2. Thermodynamic of Second Phase Formation

To further study the formation mechanism of the intermetallic compound phase, the formation enthalpy and binding energy of each metal mesophase (elemental Al, Zn, Mg and Cu, as well as single Al, Zn, Mg and Cu atoms in the ground state) was calculated to determine the formation enthalpy and binding energy of the intermediate phases of Al, Zn, Mg and Cu. The formation enthalpy (Δ*H*) can be calculated according to Equation (2) [38,39]:(2)$$\Delta H_{A_{n_{1}}B_{n_{2}}C_{n_{3}}}=\frac{1}{n_{1}+n_{2}+n_{3}}\left[E_{t}^{A_{n_{1}}B_{n_{2}}C_{n_{3}}}-n_{1}E_{S}^{A}-n_{2}E_{S}^{B}-n_{3}E_{S}^{C}\right]$$
where *E*_t_ is the total energy calculated at *T* = 0 K; and $E_{S}^{A}$, $E_{S}^{B}$, and $E_{S}^{C}$ are the energies per atom of bulks A, B, and C, respectively.

The binding energies can be calculated by using Equation (3) [40]:(3)$$E_{b}^{A_{n_{1}}B_{n_{2}}C_{n_{3}}}=\frac{1}{n_{1}+n_{2}+n_{3}}\left[E_{t}^{A_{n_{1}}B_{n_{2}}C_{n_{3}}}-n_{1}E_{a}^{A}-n_{2}E_{a}^{B}-n_{3}E_{a}^{C}\right]$$
where $E_{a}^{A}$, $E_{a}^{B}$, and $E_{a}^{C}$ are the energies per atom of free atoms A, B and C, respectively.

The calculated formation enthalpy and binding energy of the intermediate phase were shown in Table 4.

The results of thermodynamic calculations after sintering were shown in Table 4. The second-phase of Al_2_Cu formed in the sintering process owing to its maximum binding energy and stability. As shown in Figure 6 and Figure 10, a certain amount of micron-sized Al_2_Cu phase formed in SPS samples. The mesophase Al_2_CuMg might have formed during the sintering process, however, the mesophase Al_2_CuMg was unsteady that can easily decompose or convert into Al_2_Cu. The formation enthalpy of the MgZn_2_ phase was the lowest, the formation enthalpy temperature of the MgZn_2_ phase in Al-Zn-Mg alloy is usually 350 °C, which is lower than the sintering temperature [41]. During the cooling processes after sintering, Mg and Zn atoms reacted to form a second phase of MgZn_2_ with size of nanometer (Figure 10) because of the lower diffusion distance of Mg and Zn atoms at a lower reaction temperature [42,43].

#### 3.2.3. Mechanical Properties of SPS Samples

Figure 11 shows the compressive properties and Vickers microhardness of the sintered samples. It is clear from the compressive stress-strain curves in Figure 11a that the compressive strength of the SPS samples was enhanced with the increase of milling time. Figure 11b shows that the compressive yield strength was increased first and then decreased slightly. The decreasing can be attributed to the increasing of grain size [44]. Figure 11c shows that Vickers’ microhardness was also enhanced with the increase of the milling time. On the whole, S30 behaved superior comprehensive properties, with yield and ultimate strength of ~853 MPa and ~908 MPa, respectively, ~8.1% engineering strain and ~235 HV Vickers microhardness.

Samples sintered for milling time of 20 h and 30 h were selected as fracture morphologies analysis, the results were shown in Figure 12. It shows that the fracture morphology of two samples are relatively flat, and exhibit typical brittle fracture. The fracture in S20 (Figure 12a,b) has an irregular cracked surface, distinct holes, and no clear necking region. The fracture morphology of S30 has distinct necking regions, however, there are no obvious holes, as shown in Figure 12c. Figure 12d shows that the crack grows from the second-phase particles. Therefore, the pores and second-phase particles might have been the source of the cracks.

### 3.3. Strengthening Mechanism

The above experimental results demonstrate that the mechanical properties of the Al-Zn-Mg-Cu alloy prepared by the MA-SPS method are remarkably improved. The reinforcing mechanisms of Al-Zn-Mg-Cu alloy prepared by the MA-SPS method are fine-grained strengthening, dislocation strengthening, Orowan strengthening and solid-solution strengthening [45,46].

#### 3.3.1. Grain Boundary Strengthening

The Hall–Petch relationship [47] is a theoretical model for calculating the effect of grain refinement on yield strength and is provided by Equation (4):(4)$$\sigma =\sigma_{0}+Kd^{-1/2}$$
where σ is the yield strength, in MPa; $\sigma_{0}$ is the lattice frictional force to be overcome to move a single dislocation, in MPa; *d* is the average grain size; *K* is the material-dependent constant, in MPa·m^1/2^.

For the Al-Zn-Mg-Cu alloy, $\sigma_{0}$ is ~16 MPa [48], *K* is ~0.12 MPa·m^1/2^, and the value of *d* was 44–78 nm. The yield strength of S30 was calculated as 445–588 MPa.

#### 3.3.2. Dislocation Strengthening

The interaction between dislocation and grain boundary or secondary phase particles or solute atom increases the deformation resistance, thereby, improves the strength of alloys. The relation between dislocation density and yield strength can be calculated by Taylor’s equation [49]: (5)$$\Delta \sigma_{D}=K_{D}Gb\sqrt{\rho }$$
where *K_D_* is a constant, approximately ~0.2; and *G* is the shear modulus of the matrix. For the Al-Zn-Mg-Cu alloy, *G* is approximately ~26.9 GPa. Here, *b* is the Burgers vector of the matrix, ~0.286 nm, and *ρ* is the dislocation density. Arsenault et al. claimed that newly formed dislocation density near the enhanced sample can be expressed as in Equation (6) [50]:(6)$$\rho =\frac{2\sqrt{3}\varepsilon }{db}$$
where *d* is the average diameter of the crystallites, nm; *ε* is lattice distortion which can be obtained by the XRD analysis in Figure 6 and Table 3. For S30, micro strain *ε* = 0.075, and *b* is the Burgers’ vector which is ~0.286 nm. Based on these parameters, the $\Delta \sigma_{D}$ is estimated to be 194.1 MPa by combining Equations (5) and (6).

#### 3.3.3. Solid-Solution Strengthening

For the Al-Zn-Mg-Cu alloys, Zn, Mg and Cu are solute atoms that increase the lattice distortion of the α-Al matrix and cause dislocations, thereby increasing the strength and hardness of the alloys. Usually, solid-solution strengthening can be estimated by Equation (7) [51]:(7)$$\Delta \sigma_{SS}=MGb\varepsilon_{}^{\frac{3}{2}}\sqrt{c}$$

For the Al-Zn-Mg-Cu alloys, *M* is the mean orientation factor, and is ~3.06; *G* is shear modulus, and is ~26.9 GPa; *b* is Burgers’ vector, and is ~0.286 nm; *ε* is micro strain. For S30, *ε* = 0.075; for nanostructured materials. *c* is the correction factor from 0.5 to 1.

Assuming that all solute atoms are dissolved in the matrix in atomic form, the value of $\Delta \sigma_{SS}$ was ~82 MPa. However, in the sintered sample, numerous Zn, Mg and Cu atoms were precipitated in the form of the second phase. Thus, the strength contribution of solid solution strengthening is much lower than 82 MPa, and it is very limited strengthening for Al-Zn-Mg-Cu alloys prepared by the MA-SPS method.

#### 3.3.4. Orowan Strengthening

The closely spaced hard particles might cause the resistance to the passing dislocation, which is called Orowan strengthening. When the similar particles are encountered during dislocation motion, the lines of dislocation are required bypassing the particles and form an Orowan ring around the particles.

In the Orowan model, the increase in the yield strength of composites is provided by Equation (8) [52]:(8)$$\Delta \sigma_{Orowan}=M\frac{0.4Gb}{\lambda_{p}\pi \sqrt{1-\nu }}\ln\frac{d_{p}}{2b}$$
where *M* is the mean orientation factor, and is ~3.06; *G* is shear modulus, and is ~26.9 GPa; *b* is Burgers’ vector, and is ~0.286 nm; *ν* is poisson ratio, and is ~0.33. *d_p_* is particle diameter, nm; and *λ* is particle spacing, nm. Here, *λ* is related to Equation (9) [52]:(9)$$\lambda \approx d_{p}\left[{\left(\frac{1}{2V_{p}}\right)}^{\frac{1}{3}}-1\right]$$
where *V_p_* is the volume fraction of the particles in the composite.

It is assumed that the Cu, Mg, and Zn atoms were all precipitated into second-phase particles Al_2_Cu and MgZn_2_ during the sintering process, the maximum $\Delta \sigma_{O\mathrm{rowan}}$ was 450 MPa. However, not all Cu, Mg and Zn atoms were precipitated as second-phase particles; thus, $\Delta \sigma_{Orowan}$ was slightly less than 450 MPa.

Based on the calculations and analysis above, it is concluded that the high strength of the nanocrystalline bulk Al-7Zn-2.5Mg-2.5Cu alloy is attributed to fine-grained strengthening, while dislocation strengthening and Orowan strengthening due to the precipitated second phase particles are the next. The solid-solution strengthening plays a negligible role.

## 4. Conclusions

The bulk nanocrystalline Al-7Zn-2.5Mg-2.5Cu alloy was prepared by mechanical alloying and SPS sintering using metal powders such as Al, Zn, Mg and Cu. The changes of the powders’ morphology and crystal structure during the ball milling process, as well as the microstructure formation and mechanical properties of the sintered sample, were studied. The following conclusions could be obtained:(1)With the extension of the milling time, the Zn, Mg and Cu alloy elements are gradually dissolved in α-Al. When the ball milling time is 30 h, complete solid solution is achieved, and no intermetallic phase is formed. The ball-milled Al powder undergoes flattening, crushing and welding processes. When the ball milling time is 30 h, the powder particle size is 2–5 μm. The α-Al grain size was 23.2 nm. The lattice distortion was 0.156% causing by the solid solution of the alloying elements.(2)The grain growth of the ball milled powder occurred during the SPS sintering process. The grain size after ball milling for 30 h was increased by 53.5 nm from the powder 23.2 nm. Due to the desolvation of the supersaturated solid solution, micro-sized Al_2_Cu is precipitated at the α-Al grain boundary, and nano-size MgZn_2_ is precipitated in the α-Al crystal.(3)With the increase in ball milling duration, the compressive strength, yield strength and Vickers microhardness of the sintered samples increased gradually but the engineering strain decreased. The compressive strength, yield strength, Vickers microhardness and engineering strain after ball milling for 30 h were ~908 MPa, ~853 MPa, ~235 HV and ~8.1%, respectively. (4)In the nanocrystalline bulk Al-7Zn-2.5Mg-2.5Cu alloy prepared by the MA-SPS method, the high strength of the nanocrystalline bulk Al-7Zn-2.5Mg-2.5Cu alloy was attributed to fine-grained strengthening, while the dislocation strengthening and Orowan strengthening due to the precipitated second phase particles are the next.


## Figures and Tables

**Figure 1 materials-12-01255-f001:**
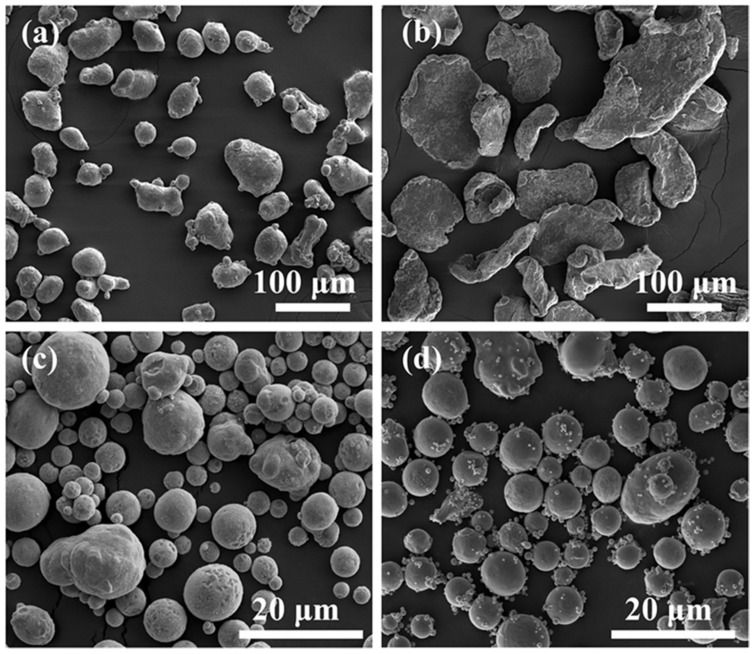
Scanning electron microscope (SEM) micrographs of the original powder (**a**) Al, (**b**) Mg, (**c**) Zn and (**d**) Cu powder.

**Figure 2 materials-12-01255-f002:**
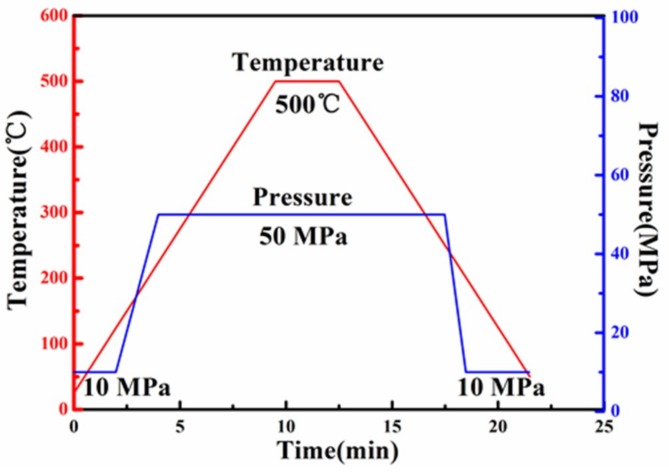
Diagram of variations in temperature and pressure during sintering.

**Figure 3 materials-12-01255-f003:**
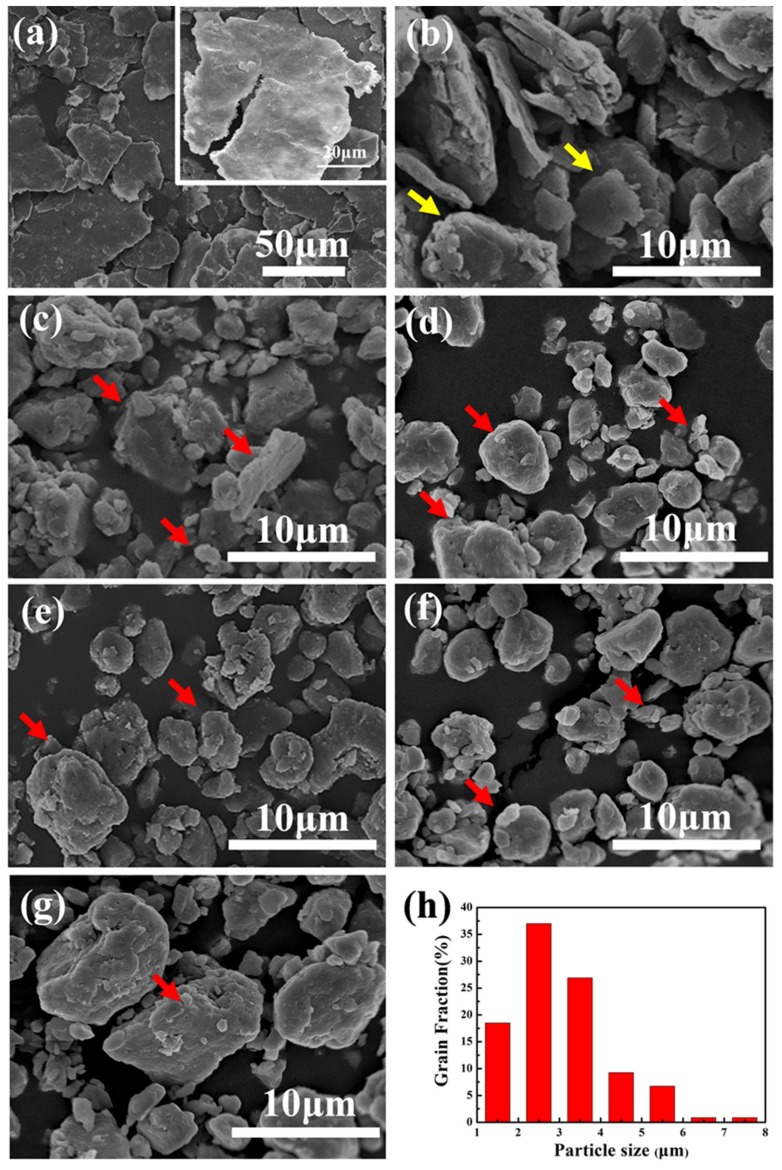
Morphologies of Al-7Zn-2.5Cu-2.5Mg milled powders with different ball milling time (**a**) P5, (**b**) P10, (**c**) P15, (**d**) P20, (**e**) P25, (**f**) P30, (**g**) P40 and (**h**) histogram of P30 particles size distribution.

**Figure 4 materials-12-01255-f004:**
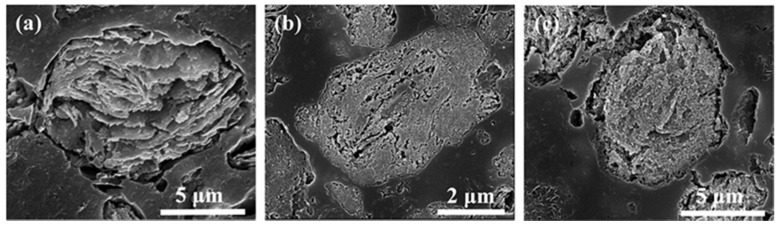
Cross-section morphologies of Al-7Zn-2.5Cu-2.5Mg milled powders at different ball milling time (**a**) P10, (**b**) P20 and (**c**) P30.

**Figure 5 materials-12-01255-f005:**
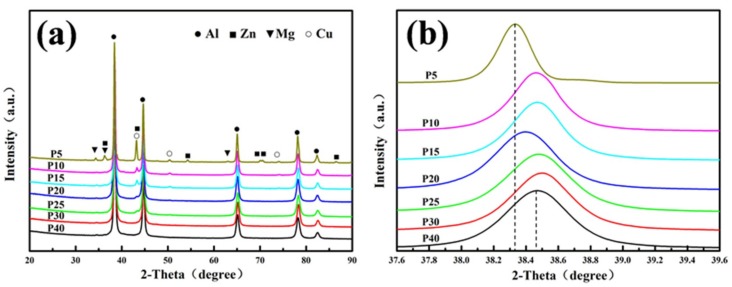
X-ray diffraction (XRD) spectrum of Al-7Zn-2.5Mg-2.5Cu alloy powders after different ball milling time (**a**) the whole patterns and (**b**) localized peak patterns.

**Figure 6 materials-12-01255-f006:**
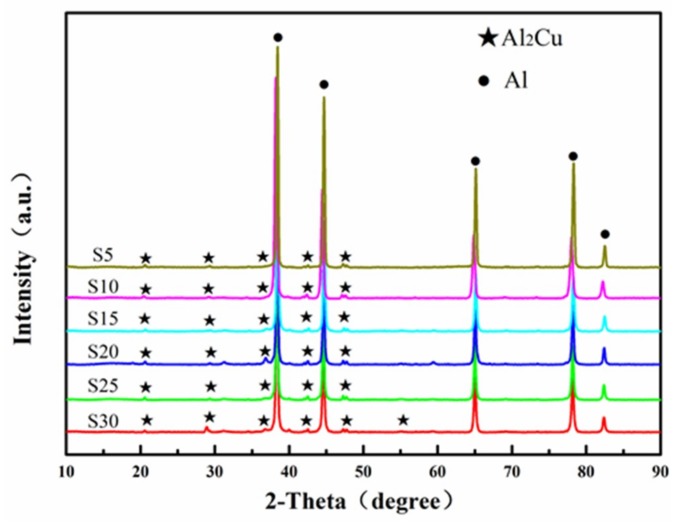
XRD patterns of sintered samples with different ball milling time.

**Figure 7 materials-12-01255-f007:**
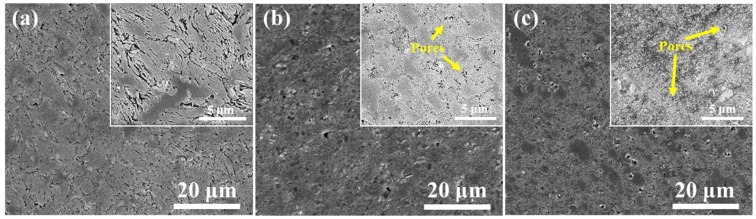
SEM micrographs of sintered samples with different ball milling time (**a**) S10, (**b**) S20 and (**c**) S30.

**Figure 8 materials-12-01255-f008:**
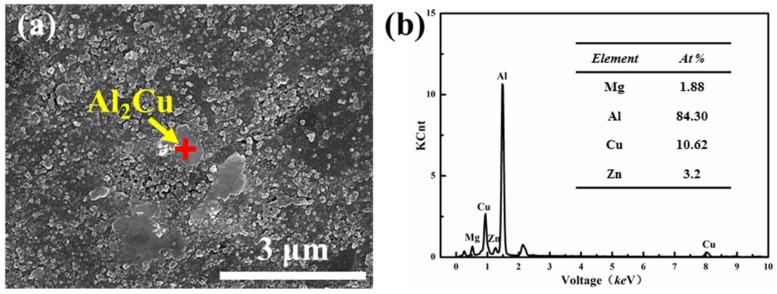
The microstructure of S30 sample (**a**) SEM micrograph and (**b**) energy-dispersive spectroscopy (EDS) analysis of secondary phase.

**Figure 9 materials-12-01255-f009:**
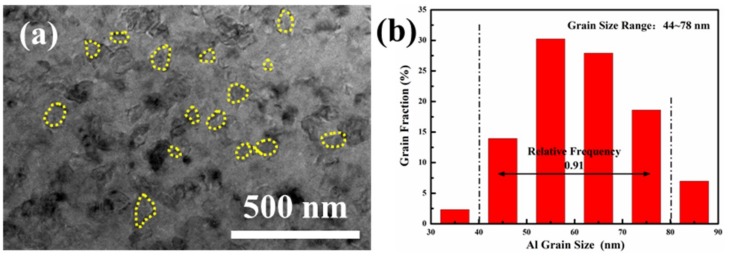
(**a**) S30 transmission electron microscopy (TEM) micrograph and (**b**) Distribution histogram of aluminum grain size.

**Figure 10 materials-12-01255-f010:**
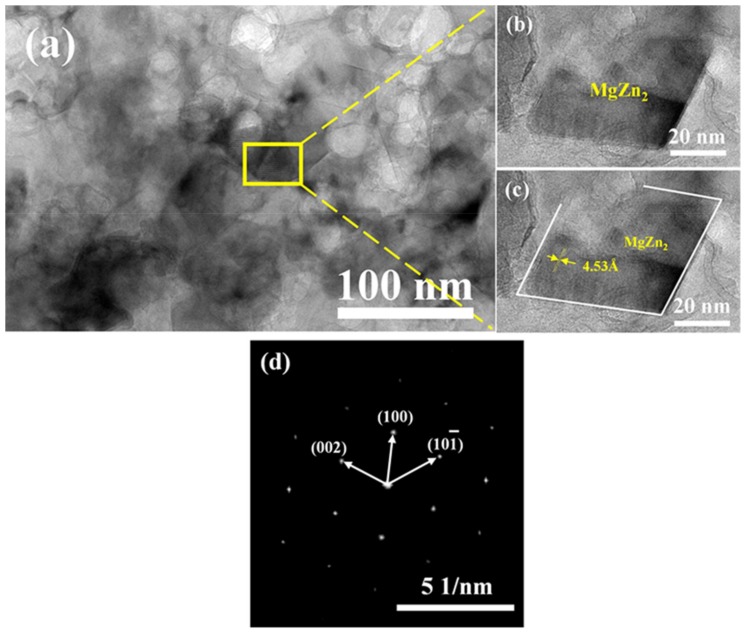
TEM analysis of S30 samples (**a**) S30 TEM micrograph, (**b**,**c**) HRTEM micrograph of MgZn_2_ nanoparticle and (**d**) Fast Fourier Transform (FFT) image of MgZn_2_.

**Figure 11 materials-12-01255-f011:**
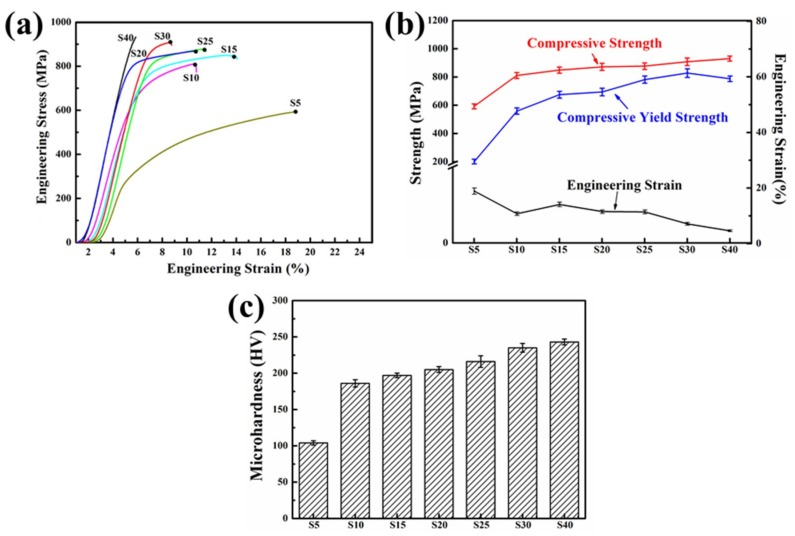
Sintered samples for different ball milling time (**a**) compressive stress-strain curves, (**b**) values of strength and strain and (**c**) Vickers microhardness.

**Figure 12 materials-12-01255-f012:**
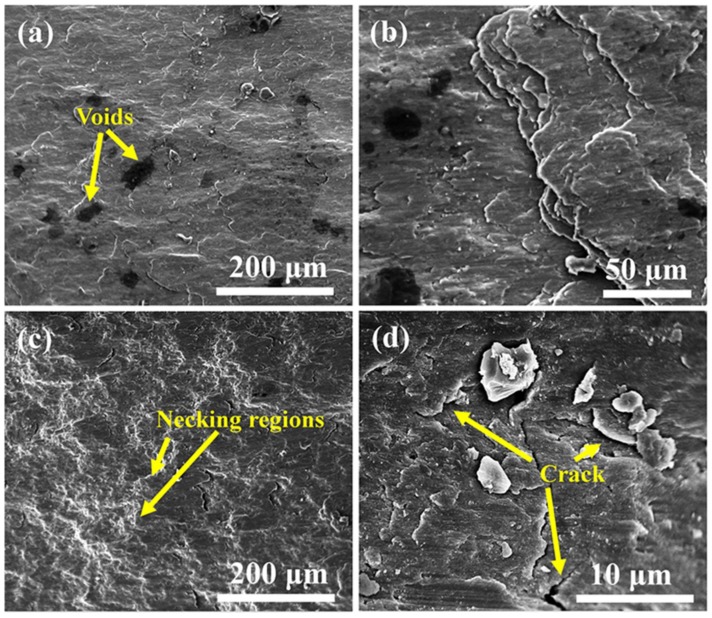
Fracture surface of samples tested under compression: (**a**,**b**) S20 and (**c**,**d**) S30.

**Table 1 materials-12-01255-t001:** Specifications of raw materials.

Elements	Al	Zn	Mg	Cu
**Mesh size (μm)**	38–75	4–23	75–150	3–20
**Purity (%)**	99.5	99.99	99	99.99

**Table 2 materials-12-01255-t002:** Grain size and microscopic strain with different ball milling time

Sample	Grain Size *d* (nm)	Micro Strain *ε* (%)
P5	58.9 ± 3.9	0.053 ± 0.0083
P10	49.1 ± 1.9	0.138 ± 0.0069
P15	51.1 ± 2.4	0.177 ± 0.0076
P20	28.9 ± 0.6	0.135 ± 0.0075
P25	28.0 ± 0.7	0.164 ± 0.0074
P30	22.3 ± 0.6	0.156 ± 0.0072
P40	23.0 ± 0.5	0.138 ± 0.0084

**Table 3 materials-12-01255-t003:** Grain sizes and micro strains of sintered samples for different ball milling time.

Sample	S20	S25	S30	S40
Grain size *d* (nm)	73.5 ± 3.3	58.5 ± 2.1	53.5 ± 2.0	60.5 ± 2.2
Micro strain *ε* (%)	0.03 ± 0.006	0.05 ± 0.005	0.075 ± 0.0056	0.085 ± 0.0045

**Table 4 materials-12-01255-t004:** Formation enthalpy Δ*H* and binding energy *E_b_* of metal intermediate phase.

Phase	MgZn_2_ (*η*)	Al_2_CuMg (*S*)	Al_2_Cu (*θ*)
Δ*H* (kJ/mol)	−10.36	−6.645	−3.86
*E_b_* (kJ/mol)	142.06	319.7	358.5

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
