# Peer review of "Microstructure and Mechanical Properties of Nanocrystalline Al-Zn-Mg-Cu Alloy Prepared by Mechanical Alloying and Spark Plasma Sintering"

_materials, 2019, doi:10.3390/ma12081255_

Round 1
Reviewer 1 Report
This paper present the preparation, structural studies and mechanical properties of an Al-based alloy produced by mechanical alloying and Spark Plasma Sintering (SPS). The paper is written in a good scientific language, which requires only minor polishing before publication. The authors obtained a mechanically strong Al-based alloy from elemental powders of Al and alloying additives of metals by conducting ball miling and fast consolidation via SPS. I think the paper can be published after minor revision.
Please use "grain boundary strengthening" instead of "fine-grained strengthening".
In the Abstract, the goal of the study should be given.
In Fig.5 (b), the position of the peak changes in a non-monotonous manner with the milling time. Please make a comment why.
The grain size should be rounded up to nanometers. It hardly makes sense to give values with decimal digits.
Please revise the caption to Fig.12 (fracture surface of samples tested under compression).
Line 351: "c varies from 1/2 to one" - what is "c"?
Did you test the synthesized alloy in tension?
Author Response
Dear editor and reviewers:
Thank you for your letter and for the reviewers’ comments concerning our manuscript entitled “Microstructure and mechanical properties of nanocrystalline Al-Zn-Mg-Cu alloy prepared by mechanical alloying and spark plasma sintering”, Manuscript ID: Materials-481925. Those comments are all valuable and very helpful for revising and improving our paper. We have studied comments carefully and have made corrections which we hope meet with approval. Revised portion are marked in yellow in the paper. The response to the reviewer’ comments is as following:
Point 1: Please use "grain boundary strengthening" instead of "fine-grained strengthening".
Response 1: “Fine-grained strengthening” has been revised into “Grain boundary strengthening.” in this manuscript.
Point 2: In the Abstract, the goal of the study should be given.
Response 2: Thanks for your suggestions. This goal of the study was given in the Abstract.
Point 3: In Fig.5 (b), the position of the peak changes in a non-monotonous manner with the milling time. Please make a comment why.
Response 3: Since the atomic radius of Zn (0.134 nm) and Cu (0.128 nm) are smaller than that of Al (0.143 nm), the peak of Al is shifted to high angle while Zn and Cu atoms are dissolved into Al matrix. Since the atomic radius of Mg (0.160 nm) is larger than that of Al, the peak of Al is shifted to low angle while Mg atoms are dissolved into Al matrix. The atoms of Zn, Cu and Mg are disordered in solid solution to Al matrix. So the position of the peak changes in a non-monotonous manner with the milling time.
Point 4: The grain size should be rounded up to nanometers. It hardly makes sense to give values with decimal digits.
Response 4: Thanks for your suggestions. The values of particle size have decimal digits calculated by Williamson–Hall method in Jade 6.0 software. The grain size also has measurement error. The measurement error needs to be described with decimal digits.
So we gave values of grain size with decimal digits.
Point 5: Please revise the caption to Fig.12 (fracture surface of samples tested under compression).
Response 5: The caption of Fig.12 has been revised into fracture surface of samples tested under compression.
Point 6: Line 351: "c varies from 1/2 to one" - what is "c"?
Response 6: c is the correction factor. This has been revised in the manuscript.
Point 7: Did you test the synthesized alloy in tension?
Response 7: SPS samples are too small to be processed into standard tensile samples.
The revised paper has been resubmitted to your journal. We look forward to your positive response.
Yours sincerely,
Jingfan Cheng
Apr 12, 2019

Reviewer 2 Report
The paper deals with the microstructure and mechanical properties of an Al-Zn-Mg-Cu alloy. Powders were prepared by mechanical alloying and bulk samples were obtained by spark plasma sintering. The paper is interesting but needs a few adjustments to be ready for publication.
Main comments are listed as follows:
- The “Introduction” section is written with a poor English form, please update it.
- Figure 1 caption: please type “SEM images of the original powders” and then list only the letters and the elements.
- Milling time from 5 hours to 40 have been reported, with steps of 5 hours between each other, except for the 30-40 interval: my question is why there are no results for the P35 and S35 samples? Please comment on this.
- Lines 122-127: please indicate the normative reference for mechanical testing at least.
- Line 146: please put the references straight after the “recent studies”.
- Lines 190-191: the same sentence is repeated twice, please change it.
- Line 195: please delete the dots between equation and number.
- Table 2: Please update the errors and write them in a better form.
- Table 2: the grain size has a measurement error which decreases while increasing the milling time: can the authors comment on this?
- Lines 236-239: the authors claim that the pores in Figure 7 are reduced in size from Figure 7(b) to (c): this result is not very clear from the figure, I would suggest including micrographs taken at higher magnification to highlight the change in the pores dimensions.
- Please, put the full name followed by the acronym when an acronym is used for the first time.
- Line 274: consideration about the mechanical properties should not be reported here, since the authors have not yet presented the results concerning the mechanical tests.
- Please, explain clearly all the terms of the equations.
- Table 4: authors report about a mesophase Al2CuMg, which has never been introduced in the paper: where was it observed? Please comment on this.
- Line 313: please type (…) instead of [….].
- Lines 362-363: please change the size of the text.
Author Response
Dear editor and reviewers:
Thank you for your letter and for the reviewers’ comments concerning our manuscript entitled “Microstructure and mechanical properties of nanocrystalline Al-Zn-Mg-Cu alloy prepared by mechanical alloying and spark plasma sintering”, Manuscript ID: Materials-481925. Those comments are all valuable and very helpful for revising and improving our paper. We have studied comments carefully and have made corrections which we hope meet with approval. Revised portion are marked in yellow in the paper. The response to the reviewer’ comments is as following:
Point 1: The “Introduction” section is written with a poor English form, please update it.
Response 1: The introduction has been revised.
Point 2: Figure 1 caption: please type “SEM images of the original powders” and then list only the letters and the elements.
Response 2: Thanks for your suggestions. This has been modified in the revised manuscript.
Point 3: Milling time from 5 hours to 40 have been reported, with steps of 5 hours between each other, except for the 30-40 interval: my question is why there are no results for the P35 and S35 samples? Please comment on this.
Response 3: Thanks for your suggestions. Based on the recent research, the influence of ball milling time on grain size became small when the ball milling time arrived at the stage of dynamic equilibrium. Therefore, after 30 h milled, we directly take 40 h to better reflect the change of particle size.
Point 4: Lines 122-127: please indicate the normative reference for mechanical testing at least.
Response 4: Thanks for your suggestions. The compression test should refer Metallic materials - Compression test method at room temperature (GB/T 7314-2017) using a Shimadzu AG-100 KN machine at a speed of 0.2 mm/min. The Vickers microhardness test should refer Metallic materials - Vickers hardness test. (GB/T 4340-2009). This has been modified in the revised manuscript
Point 5: Line 146: please put the references straight after the “recent studies”.
Response 5: The references has been put straight after the “recent studies”.
Point 6: Lines 190-191: the same sentence is repeated twice, please change it.
Response 6: The same sentence has been revised in the manuscript.
Point 7: Line 195: please delete the dots between equation and number.
Response 7: The dots has been deleted in the manuscript.
Point 8: Table 2: Please update the errors and write them in a better form.
Response 8: Thanks for your suggestions. This has been modified in the revised manuscript.
Point 9: Table 2: the grain size has a measurement error which decreases while increasing the milling time: can the authors comment on this?
Response 9: The measurement error of grain size in Table 2 decreases with the increase of ball grinding time, indicating that with the increase of ball grinding time, the size of grains tends to be more uniform.
Point 10: Lines 236-239: the authors claim that the pores in Figure 7 are reduced in size from Figure 7(b) to (c): this result is not very clear from the figures, I would suggest including micrographs taken at higher magnification to highlight the change in the pores dimensions.
Response 10: Thanks for your suggestions. We have added the high magnification micrographs in the form of illustrations. This has been modified in the revised manuscript.
Figure 7. SEM micrographs of sintered samples with different ball milling time (a) S10, (b) S20 and (c) S30.
Point 11: Please, put the full name followed by the acronym when an acronym is used for the first time.
Response 11: Thanks for your suggestions. This has been modified in the revised manuscript.
Point 12: Line 274: consideration about the mechanical properties should not be reported here, since the authors have not yet presented the results concerning the mechanical tests.
Response 12: Thanks for your suggestions. The relevant descriptions have been revised in the manuscript.
Point 13: Please, explain clearly all the terms of the equations.
Response 13: Thanks for your suggestions. This has been modified in the revised manuscript.
Point 14: Table 4: authors report about a mesophase Al2CuMg, which has never been introduced in the paper: where was it observed? Please comment on this.
Response 14: Thanks for your suggestions. The mesophase Al2CuMg might have formed during the sintering process. However, the mesophase was unsteady that can easily decompose or convert into Al2Cu. So The mesophase Al2CuMg was not observed in SEM or TEM micrographs.
Point 15: Line 313: please type (…) instead of [….]
Response 15: This has been revised in the manuscript.
Point 16: Lines 362-363: please change the size of the text.
Response 16: This has been revised in the manuscript.
The revised paper has been resubmitted to your journal. We look forward to your positive response.
Yours sincerely,
Jingfan Cheng
Apr 12, 2019

Reviewer 3 Report
Dear Authors,
I have read your manuscript carefully and I would say that this manuscript would be very interesting for readers.Combining mechanical alloying and spark plasma sintering is a well-known and promising route to produce advanced materials. What is more, aluminum alloys, especially nanostructured ones, are becoming increasingly more attractive for many branches of industry. This means that your results fit in an actual trend in materials science. The objectives of the study are clearly defined. The introduction provides a good, generalized background of the topic. The methods and results are clearly explained and are presented in an appropriate format. The figures and tables show essential data; some of the data are also summarized in the text. I do not think any additional graphics are necessary. The cited literature is relevant to the study and balanced. Unfortunately, this manuscript has not been prepared well and cannot be published in its present form. It will only be possible after correcting some linguistic and editing errors. Moreover, extensive English language editing is needed by a native speaker in view of the poor English of this manuscript.
I highlighted the main remarks in yellow in the PDF version of your manuscript. To clarify:
- I recommend presenting the nomenclature of the samples (P5 – P40 and S5 – S40) at the beginning, and use it consistently until the end, not sometimes using, e.g. 5h, others S5.
- It will be enough if you show the values of strength in integers, not in hundredths.
- In many cases there is a lack of spaces, words begin with a capital letter despite the fact they do not start a new sentence, etc.
- I suggest using one style to present the weight, volume or atomic content of elements.
- In all cases, it should be SEM or TEM micrographs instead of SEM images.
- I suggest using the symbol of diameter “Ø” instead of “ø”.
- In Figure 11a, stress, not strength is presented on the Y-axis. Strength is presented only in Figure 11b.

Author Response
Dear editor and reviewers:
Thank you for your letter and for the reviewers’ comments concerning our manuscript entitled “Microstructure and mechanical properties of nanocrystalline Al-Zn-Mg-Cu alloy prepared by mechanical alloying and spark plasma sintering”, Manuscript ID: Materials-481925. Those comments are all valuable and very helpful for revising and improving our paper. We have studied comments carefully and have made corrections which we hope meet with approval. The poor English of this manuscript has been revised. Revised portion are marked in yellow in the paper. The responses to the reviewers’ comments are as following:
Point 1: I recommend presenting the nomenclature of the samples (P5 – P40 and S5 – S40) at the beginning, and use it consistently until the end, not sometimes using, e.g. 5h, others S5.
Response 1: Thanks for your suggestions. The nomenclature of the samples has been uniformed and used consistently. This has been modified in the revised manuscript.
Point 2: It will be enough if you show the values of strength in integers, not in hundredths.
Response 2: The values of strength has been revised in integers in all this manuscript.
Point 3: In many cases there is a lack of spaces, words begin with a capital letter despite the fact they do not start a new sentence, etc.
Response 3: Thanks for your suggestions. This has been modified in the revised manuscript.
Point 4: I suggest using one style to present the weight, volume or atomic content of elements.
Response 4: Thanks for your suggestions. This has been modified in the revised manuscript. The content of elements was represented in one style. (in weight percent)
Point 5: In all cases, it should be SEM or TEM micrographs instead of SEM images.
Response 5: SEM or TEM images have been revised into SEM or TEM micrographs in this manuscript.
Point 6: I suggest using the symbol of diameter “Ø” instead of “ø”.
Response 6: The symbol of diameter “Ø” has been instead of “ø”.
Point 7: In Figure 11a, stress, not strength is presented on the Y-axis. Strength is presented only in Figure 11b.
Response 7: Thanks for your suggestions. This has been modified in the revised manuscript.
Figure 11. Sintered samples for different ball milling time (a) compressive stress-strain curves, (b) values of strength and strain and (c) Vickers microhardness.
The revised paper has been resubmitted to your journal. We look forward to your positive response.
Yours sincerely,
Jingfan Cheng
Apr 12, 2019
